# Archetypal analysis of COVID-19 in Montana, USA, March 13, 2020 to April 26, 2022

**Emily Stone**[1]*, **Sebastian Coombs**[1], **Erin Landguth**[2]

**1** Dept. of Mathematical Science, University of Montana, Missoula, MT, United States of America, **2** Center for Population Health Research, School of Public and Community Health Sciences, University of Montana, Missoula, MT, United States of America

* stone@mso.umt.edu

## Abstract

Infectious disease data can often involve complex spatial patterns intermixed with temporal trends. Archetypal Analysis is a method to mine complex spatio-temporal data, and can be used to discover the dynamics of spatial patterns. The application of Archetypal Analysis to epidemiological data is relatively new, and here we present one of the first applications on COVID-19 data from March 13, 2020 to April 26, 2022, for the counties of Montana, USA. We present three views of the data set decomposed with Archetypal Analysis. First, we evaluate the entire 56 county data set. Second, we use a mutual information calculation to remove counties whose dynamics are mainly independent from the other counties, reducing the set to 17 counties. Finally, we analyze the top ten counties in terms of population size to focus on the dynamics in the large cities in the state. For each data set, we analyze four significant disease outbreaks across Montana. Archetypal Analysis uncovers distinct spatial patterns for each outbreak and demonstrates that each has a unique trajectory across the state.

**Data Availability Statement:** Data cannot be shared publicly from the Montana Dept of Health and Human Services. Data are available from the MT DPHHS (Communicable Disease Epidemiology Section. The point of contact at DPHHS is Laura

## Introduction

The COVID-19 pandemic launched an intensive effort to understand the processes and drivers of infectious disease outbreaks, with an emphasis on improving predictions and providing information to mitigate public health threats [1]. The unprecedented collection of spatially and temporally dense disease data from across the world, along with increased computing power has made many theoretical spatio-temporal approaches now computationally tractable. Combining spatial with temporal methods allows for the investigation of both the persistence and time evolution of patterns.

Archetypal Analysis (AA) is a promising machine learning tool for epidemiological data and the analyses of the spatio-temporal spread of disease. AA decomposes the data into spatial and temporal components; the archetypes being the spatial patterns and the time dependence captured by the reconstruction coefficient time series. Cutler and Breiman introduced AA as variant of Principal Component Analysis (PCA) [2] that could capture 'archetypal patterns' in the data [3]. The archetypal patterns are convex combinations of the data points, and as such resemble the data, making interpretation much more transparent than linear decompositions

Williamson: 406-444-0064) for researchers who meet the criteria for access.

**Funding:** EL: This research was supported by the National Institute of General Medical Sciences of 428 the National Institutes of Health (NIH), United States [Award Number P20GM130418] nigms.nih. gov The funders did not play any role in study design, data collection and analysis, decision to publish, or preparation of the manuscript.

**Competing interests:** The authors have declared that no competing interests exist.

such as PCA. In turn, each data point can be constructed from a convex combination of the archetypes.

PCA provides the most efficient way to compress data, but the eigenvectors produced from PCA should not be interpreted as actual data points, but as directions in the high dimensional space being decomposed. In highly clustered data, AA will find the centroids of the clusters, representing data as convex combinations of the centroids, which means it can indicate if a data point lies between several clusters. Therefore, AA combines the strengths of other commonly used techniques for data decomposition; providing the interpretability that PCA lacks, and more flexibility than many clustering algorithms.

AA applications appear in many fields, the first spatio-temporal application being the study of cellular flames [4], and now include the analysis of weather and climate patterns [5–7], machine learning [8], market analysis [9], and biomedical and industrial engineering [10, 11]. For example, recently AA was applied to large scale patterns of seas-surface temperature to represent extreme climate variability associated with marine heatwaves [12]. Mokhtari et al. [13] used AA in one of the first applications to epidemiological data, reconstructing the spatio-temporal patterns in an influenza time series, and showed how prominent outbreaks developed in time and across space for each influenza season in the years 2010–2019, in the state of Montana.

Here, we follow the approach from Mokhtari et al. [13] and apply AA to COVID-19 county-level data in Montana, USA, from March 2020 to April 2022. We use AA to find different disease outbreaks across Montana from distinct sets of archetypes and follow these outbreaks in time with the reconstruction coefficient time series. Our goal is to further evaluate the use of AA to construct and interpret the spatial patterns for the pandemic. We apply AA to decompose the full data set (56 counties), and two reduced dimension data sets.

The sections of the paper are as follows:

Methods: the data, the AA method, mutual information explained. Results: AA analysis of the full 56 dimensional data set, AA on a set reduced using mutual information (17 dimensional), AA analysis of the set restricted to counties with large population centers (10 dimensional), to isolate the study to the spread of COVID between them. Discussion: an overview of the conclusions arrived at in the Results section.

## Methods

### COVID-19 data for Montana, USA

COVID-19 data from counties ($m = 56$; i.e., spatial attributes) in Montana, USA, was acquired courtesy of the Montana Department of Health and Human Services, from March 13, 2020—April 26, 2022. These data cover 110 full weeks or n = 776 daily case counts. The first case count was recorded on March 13, 2020, however, Montana went into lockdown and had no significant number of cases until June 2020. Therefore, we analyzed the period from June 20, 2020—April 26, 2022, resulting in $n = 676$ daily data points over 56 counties. The running weekly average of COVID-19 cases per day was smoothed using the "smoothdata" function within MATLAB (default setting of mean = 3) to reduce noise in the time series (Fig 1). Cases are per 1000 people in each county, dividing by the population size of the county in 2020 (American Community survey data from the Census Bureau). In what follows we, partition the COVID-19 pandemic for Montana empirically into four outbreaks (rise in cases followed by a decline): Initial (Jun. 2020—Mar. 2021), Early Delta (July—Aug. 2021), Delta (Sept.- Dec. 2021), and Omicron (Jan. 2021—Apr. 2022).

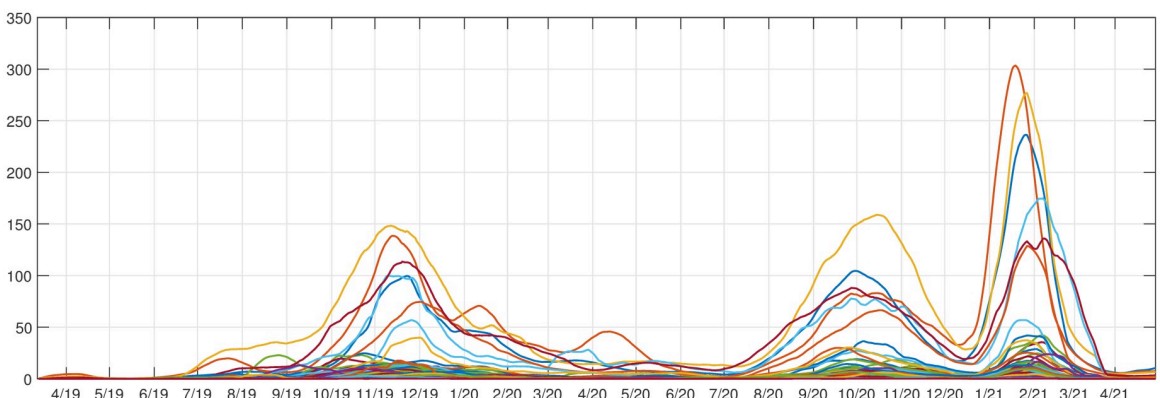

**Fig 1. COVID-19 cases in Montana, USA.** Running weekly average of COVID-19 cases plotted for all 56 Montana counties. The four outbreaks are seen here: Initial (Sept. 2020-March 2021), Early Delta (April-Aug. 2021), Delta (July—Sept. 2021), and Omicron (Jan. 2021-April 2022).

### Archetypal analysis: Mathematical formulation

Consider an $m \times n$ matrix $\mathbf{X}$, where $n$ is the number of days (676) across $m$ Montana counties (56). AA decomposes the spatio-temporal variability of $\mathbf{X}$ in a similar way to PCA but with the following underlying constraints. Given a specified value for $k$, AA identifies $m$-dimensional vectors $\mathbf{z}_1, \ldots, \mathbf{z_k}$ that best describe $k$ characteristic patterns, or archetypes, in the original data set, such that data can be represented as convex combinations (i.e., linear combinations with non-negative coefficients that sum to unity) of these archetypal patterns. The archetypes themselves are convex combination of the data points, $x_i$, $i = 1, \ldots, n$:

$$\mathbf{z}_j = \sum_{i=1}^{n} \beta_{ij} \mathbf{x}_i, \quad \beta_{ij} > 0, \forall i, j \;\; \& \;\; \sum_{i=1}^{n} \beta_{ij} = 1, \forall j. \tag{1}$$

where the $n$-dimensional vector $\boldsymbol{\beta}_j$ contains the convex weights for the $j$th archetype across all data points. The $n \times k$ matrix of all such weights is given by $\mathbf{B} = \boldsymbol{\beta}_1, \ldots, \boldsymbol{\beta}_k$. All data points can then be approximated by a convex combination of the archetypes:

$$\hat{\mathbf{x}}_i = \sum_{j=1}^{k} \alpha_{ji} \mathbf{z}_j, \quad \alpha_{ji} > 0 \;\; \forall i, j \;\; \& \;\; \sum_{j=1}^{k} \alpha_{ji} = 1 \;\; \forall i. \tag{2}$$

The convex weights, $\alpha_{ji}$ with $j = 1, \ldots, k$, sometimes referred to as mixture coefficients, range from 0 to 1, are used to reconstruct the $i$th observation across the $k$ archetypes. The $k \times n$ matrix of all such weights is given by $\mathbf{A} = \{\boldsymbol{\alpha}_1, \ldots, \boldsymbol{\alpha}_n\}$. The $k$ dimensional vectors, $\boldsymbol{\alpha}_j$'s, are like a (nonlinear) projection of the original data $\mathbf{X}$ onto $\mathbf{z}_j$, similar to PC scores in PCA. Thus the $\boldsymbol{\alpha}_j$'s are time series that determine how much of each archetype is used in reconstructing each data point.

The $m \times k$ matrix $\mathbf{Z}$ of $k$ archetypes is found by solving the optimization problem:

$$\text{argmin}_{A,B} \|\mathbf{X} - \mathbf{A}^{\mathbf{T}} \mathbf{B}^{\mathbf{T}} \mathbf{X}\|, \tag{3}$$

where $\mathbf{Z} = \mathbf{X}^{\mathbf{T}} \mathbf{B}$. $RSS = \|\mathbf{X} - \mathbf{XBA}\|$ is the residual sum of square errors, where $\|.\|$ is the spectral norm. AA seeks to find $k$ $m$-dimensional archetypes such that the $RSS$ is minimized. This approach is described in detail in [3], but can be summarized as follows: AA uses a convex least-squares method (CLSM) to estimate the coefficient $\alpha_{ji}$, subject to the constraints for

given some initial values of $\beta_{ij}$. It then finds the best $\beta_{ij}$ using CLSM, using the new $\alpha_{ji}$. This process repeats until the RSS fails to improve, or until a maximum number of iterations is reached. AA will find local minima, not necessarily the global minimum of RSS, hence using several starting $\beta_{ij}$ values to ensure a global solution is recommended. Furthermore, there is no universal method for determining the optimal value of $k$. One commonly used approach is the "elbow" criteria, where a good value of $k$ is selected by when the RSS fails to decrease with increasing number of archetypes. Since its introduction, other algorithms have been developed to find an archetypal decomposition of data. To compute the archetypes we used Matlab packages by Morten Mørup and Lars Kai Hansen [8] for computing the Principal Convex Hull [14, 15]. The AA algorithm has also been implemented in R, see [16]. See the paper [12] for a good explanation of the geometrical interpretation of the archetypes as points on the convex hull of the data, and a new algorithm for computing archetypes based on differential geometry (the manifold-based algorithm). Archetypes are presented here as color mapped counties within a state map, and were created with GeoPandas (GeoPandas.org).

It is noted in [3] that the archetypal points $\mathbf{z}$, viewed as vectors, are not orthogonal and have no natural nesting structure, i.e., as more archetypes are found, the archetypes in the smaller set can change. This is in contrast to PCA, where the set of the leading $N$ principal components are a subset of the set of the leading $M$ principal components for $M > N$. In PCA all the eigenvectors are found in a single decomposition, and the computation is fast and efficient. This is the result of the linearity of PCA, and it comes at the cost of interpretation.

We also note that, depending on how they are computed, the approximate archetypes may or may not represent extremes of the data, that is on the envelope of the convex hull. The convex hull of a data set is the smallest convex set that contains the data. The manifold-based algorithm will naturally find points on the envelope, but the alternating optimization algorithm, which minimizes the error in a convex representation of the data, may first find points representative of heavily sampled values. By the same token, the decomposition is sensitive to outliers, so extremes may be found first. This depends strongly on how the data are distributed, especially in high dimensional spaces. The error is the residual sum of squares, so if a point occurs often in a data set, the error in reproducing it will be multiplied by the number of occurrences and the algorithm will use it as an archetype, necessarily. It may or may not be an extreme in the data set. Mørup et al.'s algorithm does optimization based on projected gradient, to efficiently minimize the RSS.

## Mutual information

To reduce the dimension of the data set, we applied information-theoretic measures introduced by Shannon [17] to quantify the dependence of the count time series from different counties upon each other. For instance, if a county has a COVID-19 count time series that runs essentially independently of the other counties (as is typical of very small population counties), we could choose to remove it from the data set, thereby reducing the dimension. We determined this by calculating the mutual information between counties.

Mutual information measures the expected reduction in uncertainty about $x$ that results from learning $y$, or vice versa, where $x$ and $y$ are samples of the random variables $X$ and $Y$. This quantity can be formulated

$$I(X; Y) = H(X) + H(Y) - H(X, Y),$$

where **entropy** is defined

$$H(X) = -\sum_{x \in \mathcal{X}} p(X = x) \log_2 p(X = x) \tag{4}$$

and the **joint entropy** of two random variables $X$ and $Y$ quantifies the uncertainty of their joint distribution.

$$H(X, Y) = -\sum_{y \in \mathcal{Y}} \sum_{x \in \mathcal{X}} p(X = x, Y = y) \log_2 p(X = x, Y = y) \qquad (5)$$

Using Eqs (4) and (5), the mutual information can be rewritten

$$I(X; Y) = \sum_{x \in \mathcal{X}} \sum_{y \in \mathcal{Y}} p(x, y) \log_2 \frac{p(x|y)}{p(x)}. \qquad (6)$$

$I$ is symmetric in the variables $X$ and $Y$, i.e. $I(X; Y) = I(Y; X)$, and is zero if the random variables are independent or if the relationship between them is deterministic (nothing to be learned in either case). Note also that if $X$ is statistically correlated to $Y$, $H(X|Y)$ will be less than $H(X)$, and $I$ will be greater than 0. If $X$ is independent of $Y$, $H(X|Y) = H(X)$ and $I = 0$. If $X$ is uniquely determined by $Y$, $H(X|Y) = 0$ and $I(X; Y) = H(X)$.

In general, association measures like correlation coefficient or mutual information are used to estimate the relationships between two random variables. Correlation coefficient measures such as Pearson or Spearman entail the assumption of linear dependence. Therefore, if two random variables are associated by a nonlinear relationship these methods may fail to detect this link, or the strength will be wrongly estimated. Mutual information, however, is able to detect both linear and nonlinear dependencies, and it measures the amount of information connecting two random variables, in this case between disease cases in two Montana counties. In other words, it estimates the reduction in uncertainty about the COVID-19 activity of one county when the activity of another county is known.

## Summary of data sets and analyses

We conducted three different Archetypal Analyses. First, we used AA on the entire data set ($m$ = 56 counties). Second, we introduced MI as a systematic approach to reduce dimensionality and calculated AA on the highest mutual information data set ($m$ = 17). Third, to focus on the dynamics within and between the significant population centers, we computed AA on the counties with large populations ($m$ = 10). For each data set we constructed scree plots with the residual sum of squares error in a reconstruction (RSS) vs the number of archetypes used. These were used to choose the number of archetypes in the decomposition. Archetype sets are plotted as color-coded heat maps of counties in the state of Montana. In the reduced dimension sets, we plot the reconstruction of the data with the archetypes to illustrate the validity of the approach. We study the composition of the archetypes, and compare this with the data. The contribution of each archetype to an outbreak (Initial, Early Delta, Delta and Omicron) was then parsed out with the $\alpha$ coefficients, which allowed us to identify each with certain archetypes.

## Results

### Archetypal analysis of full data set (56 counties)

The application of AA to the data from all 56 counties of Montana is a natural place to begin our study and we include this analysis to illustrate potential problems with applying AA, and to show the complete picture of the COVID-19 epidemic in Montana. As mentioned in an earlier section, we normalized the time series by dividing by the population of each county in 2020 and multiplying it by 1000. This seems like a reasonable step, and indeed it is customary practice, but in a sparsely populated state like Montana (14 of 56 counties have population less

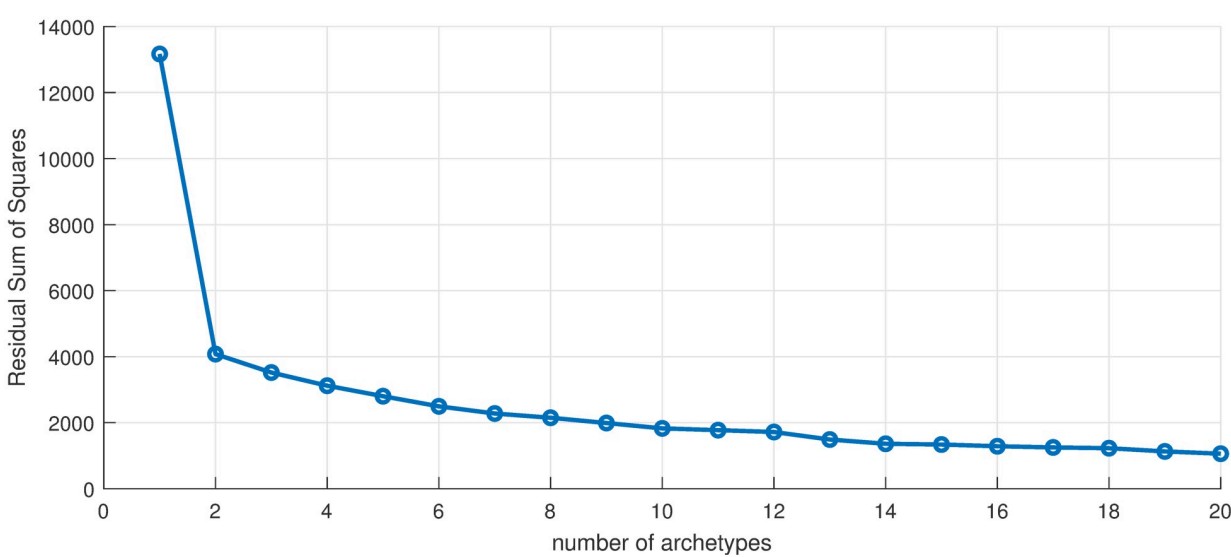

**Fig 2. Scree plot.** Residual sum of squares vs. number of archetypes in the 56 county data set. Used to select the number of archetypes in the decomposition.

than 5000, and 4 have populations less than 1000) it can lead to issues with the decomposition, as we shall demonstrate now.

We first apply archetypes to the entire truncated in time data set, which is 676 ($m$) data points in 56 ($n$) dimensions. The scree plot Fig 2) illustrates the drop-off in error as the number of archetypes is increased. Using an elbow criterion would indicate truncating to one archetype, but this is the "no-disease" or zero archetype and thus serves to turn the outbreak off and on, and would not yield any information about spatial structure. Instead, we choose $k = 10$ archetypes, which captures approximately 86% of the total variance of the data set. We are left with a 56 by 10 ($n \times k$) set of $\beta$ values, and a 10 by 676 ($k \times m$) set of $\alpha$ values.

These 10 archetypes are shown in Fig 3 (and summarized in Table 1) as color-mapped counties in Montana. Each archetypal map is a representation of the spatial features of the pandemic in a given period of time. The color is determined by the $\beta$ value for that county. We define high, medium and low counts in counties as those with $\beta$ above 1.5, between 0.75–1.5, and below 0.75, respectively.

The archetypes in this set are dominated by large counts in small population counties. When normalized by county population these have counts much higher than the largest population counties, and thus, are outliers. They would contribute a large amount to the RSS if not included in the set. In contrast, their time series are the most stochastic, and less likely to have any real predictive relationship with the other counties. Only two (numbers 8 and 10, Fig 3h and 3j) represent large counts in the largest population counties (Yellowstone, Missoula, Gallatin, Flathead, Cascade, Lewis & Clark, Ravalli, Silver Bow, Lake and Lincoln). To remove these in a systematic way, we use mutual information.

## AA on the data set with spatial dimension reduced using information theory

In sparsely populated states with a large number of counties, small population counties are numerous; 36 out of 56 in Montana have population below 10000. In decreasing order from 9391 to 434, these are: Beaverhead (at 9391 in 2020), Deer Lodge, Dawson, Stillwater,

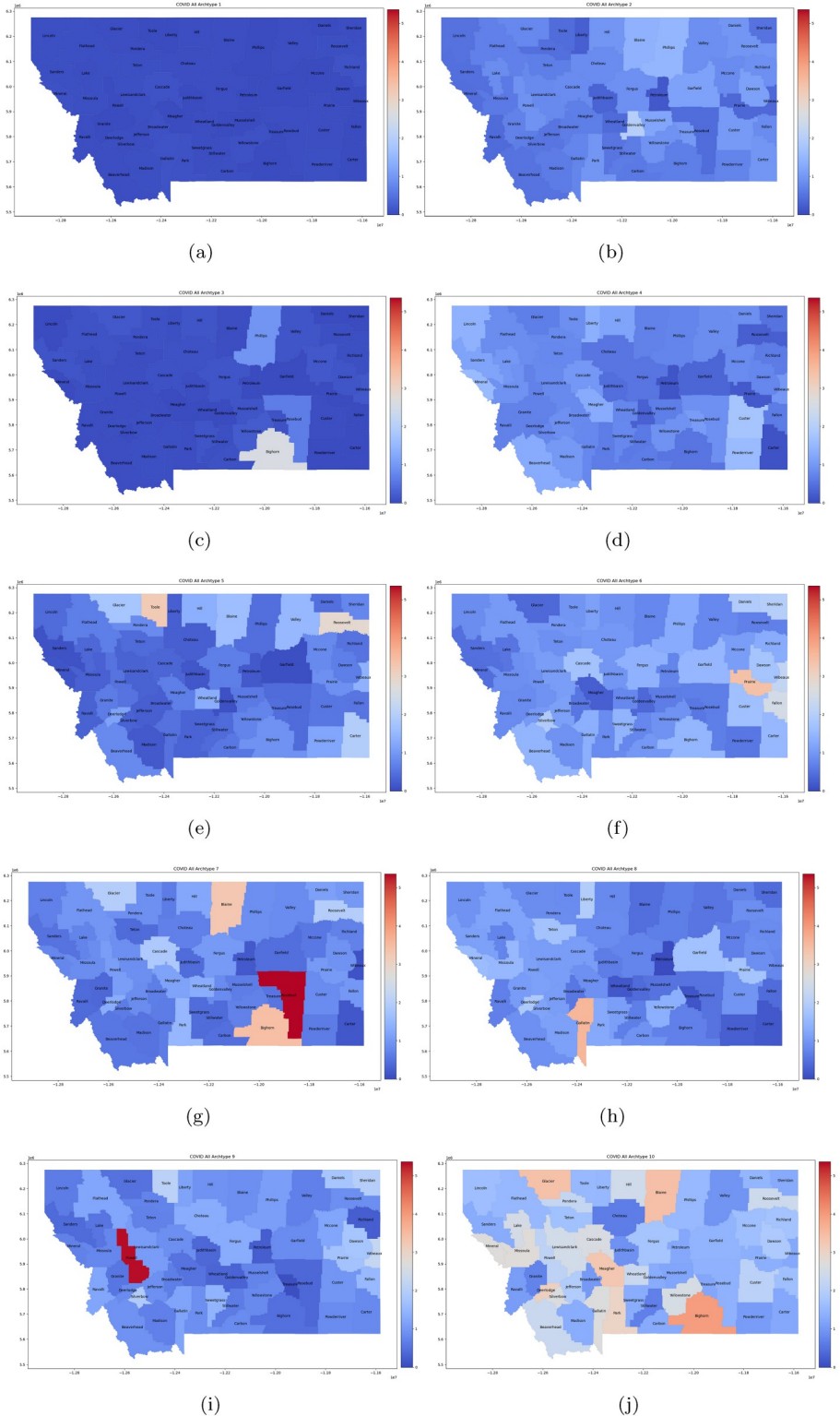

**Fig 3. Ten archetype set for all 56 counties.** Presented as $\beta$ color-coded counties in the map of Montana. The first archetype [a] is nearly zero, as it captures the "no disease" state, and acts to "turn-off" the infection/spread in each county. The rest are in order: [b] archetype 2, [c] archetype 3, [d] archetype 4, [e] archetype 5, [f] archetype 6, [g] archetype 7, [h] archetype 8, [i] archetype 9, [j] archetype 10. High $\beta$ values are considered those greater than 1.5, mid-size between 0.75 and 1.5, and below 0.75 as low. Figure created with ArcGIS: Esri, HERE, Garmin, FAO, NOAA, USGS, EPA.

**Table 1. Archetype composition-all county set.**

| Archetype | Counties in Archetype |
|---|---|
| 1 | The zero or "no disease" archetype. |
| 2 | Widespread mid-size outbreak across state, largest in small population counties. |
| 3 | Isolated larger outbreak in Big Horn county, small in Phillips, very low in the rest. |
| 4 | Widespread low to mid-size outbreak with a larger outbreaks in Ferris, Valley, and Musselshell counties, and slightly larger outbreaks in Park, Dawson, and Phillips counties. Note that these are all very low population counties. |
| 5 | Very low level counts in all counties except for ten small population counties scattered all over the state, which have mid-size outbreaks. |
| 6 | Widespread mid-size outbreaks, with larger outbreaks in small population counties, with the exception of Silverbow. |
| 7 | Low to mid-size outbreaks in all counties except Big Horn, Cascade, Glacier, Blaine, Roosevelt, and Rosebud, which have large outbreaks. The last three are very small population counties. |
| 8 | Large outbreaks in the high population counties of Yellowstone, Missoula and Gallatin, as well as the small population counties of Deer Lodge, Garfield, Liberty, Mineral, Prairie, and Teton. |
| 9 | Low to mid-size outbreaks in all counties except for nine very small counties which have very large outbreaks (Daniels, Dawson, Fallon, Powell, Sheridan, Silverbow, Sweet Grass, Toole, Wibaux). |
| 10 | Widespread large outbreaks throughout state, both in large and small population counties. |

Madison, Rosebud, Valley, Blaine, Powell, Broadwater, Teton, Pondera, Chouteau, Tolle, Musselshell, Minreal, Phillips, Sweet Grass, Sheridan, Granite, Fallon, Wheatland, Liberty, Judith Basin, Meagher, McCone, Powder River, Daniels, Carter, Prairie, Wibaux, Garfield, Golden Valley, Treasure, Petroleum (at 434 in 2020). These distort the AA by introducing stochastic outliers in the per county population data set. To determine which counties should be included in the Archetypal analysis, we computed the mutual information between all counties, and ranked counties according to their total mutual information (see also [13]).

Following the formulas in the Methods Section, we created histograms of the time series data for single counties, and joint histograms for each county with all the others. We note that the choice of bin size in these histograms will change the value of the entropy, but by choosing a fixed bin size for all the histograms, it is possible to compare the measure relative to others. Accordingly, we calculated the entropy with a uniform bin size and 30 partitions, for each

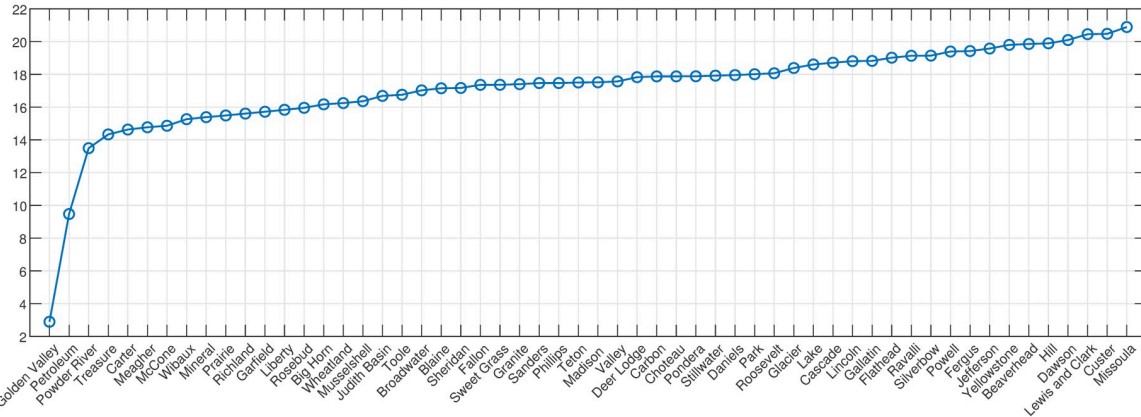

**Fig 4. Total mutual information for all 56 counties in Montana, USA.** Total mutual information (y-axis) across all counties in increasing order.

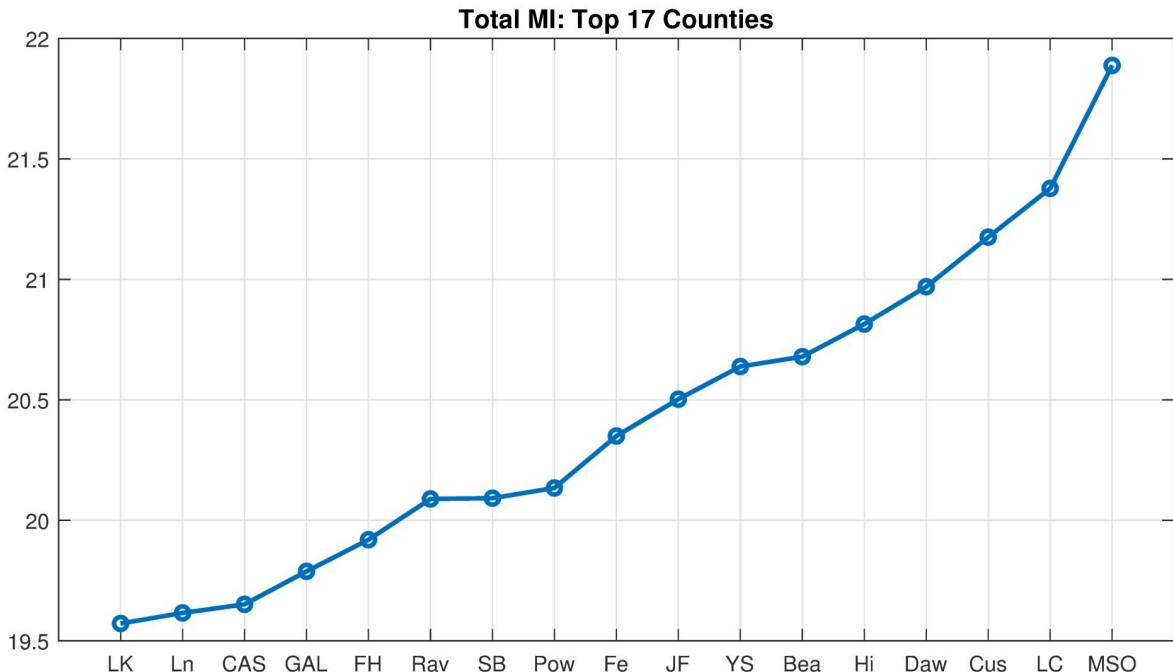

**Fig 5. Total mutual information for the largest MI counties.** Total mutual information in the top 17 total MI counties in increasing order.

county with respect to all others, creating a 56 by 56 matrix. To determine which counties have the highest MI in total, the MI row(column) for each county is summed and ordered, to create the graph shown in Fig 4.

We chose the 17 top MI counties (see Figs 5 and 6) to analyze. Taking 17 includes all the large population counties, and the smaller population counties that have the largest MI. The large population counties, with population in 2023, in alphabetical order, are: Cascade (84864), Flathead (111814), Gallatin (124857), Lake (32853), Lewis & Clark (73832), Lincoln (21525), Missoula (121041), Ravalli (47298), Silverbow (36068) and Yellowstone (169852). The

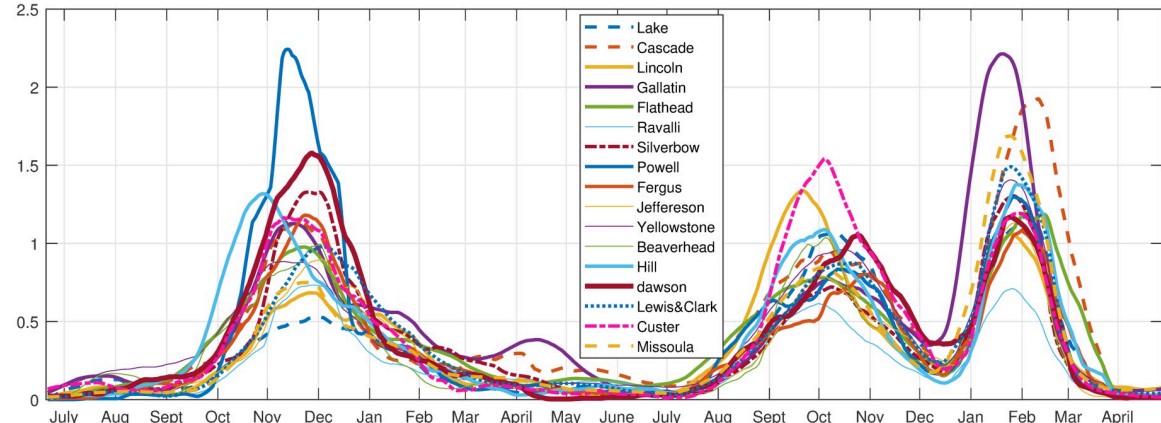

**Fig 6. Time series of COVID-19 case counts.** COVID-19 cases for the 17 largest total MI counties in Montana, USA.

**Table 2. Top 17 MI counties grouped into geographical regions.**

| Region | Counties in Region (Large Market Town, if one exists) |
|---|---|
| Northwest | Flathead (Kalispell), Lake (Polson), Lincoln (Libby) |
| North Central | Cascade (Great Falls), Fergus, Hill, Lewis & Clark (Helena, state capital) |
| Southwest | Beaverhead, Missoula (Missoula), Powell, Ravalli (Hamilton) |
| South | Gallatin (Bozeman), Jefferson, Silverbow (Butte), Yellowstone (Billings) |
| East | Custer, Dawson |

smaller population counties in alphabetical order are: Beaverhead (9719), Custer (12032), Dawson (8830), Fergus (11663), Hill (16068), Jefferson (12826), and Powell (7051). These counties can be grouped into rough geographic sub-regions. See Table 2.

Next, AA was used to decompose the COVID-19 counts into a limited number of spatial patterns, and the $\alpha$ time series. Computing archetypal sets with increasing cardinality gives the scree plot in Fig 7. Note that the largest drop in RSS occurs after the first archetype, which, as mentioned earlier, captures the "no disease" state. Beyond that, the RSS declines more slowly to near zero for numbers larger than about 15, because the set is 17 dimensional.

Using an elbow criterion on the scree plot to determine a threshold, we chose 9 archetypes for the decomposition. See Fig 8 for color heat maps of the archetypes, which are summarized in Table 3. We can check the validity of the 9 archetype decomposition of the data by comparing the time series for each county, and its reconstruction with archetypes. See Fig 9. The visible error is generally negligible, because with 9 archetypes, 99% of the variance of the data set is captured.

The sum of the $\alpha$ time series is equal to 1 for each data point, so one archetype can be ignored without loss of generality; its $\alpha$ time series can be calculated from the remaining. We chose to ignore the zero archetype when analyzing the outbreaks, as it can be inferred from the other $\alpha$'s.

The archetypes separate out clusters of counties which have simultaneous outbreaks (spatial), and the $\alpha$'s give the sequence in which the outbreaks occur (temporal). The $\alpha$ time series indicate when each archetype is active.

Fig 10 illustrates how the each of the 9 archetypes dominate uniquely during the different outbreaks, and is summarized in Table 4. The Initial outbreak is captured by archetypes 7, 5

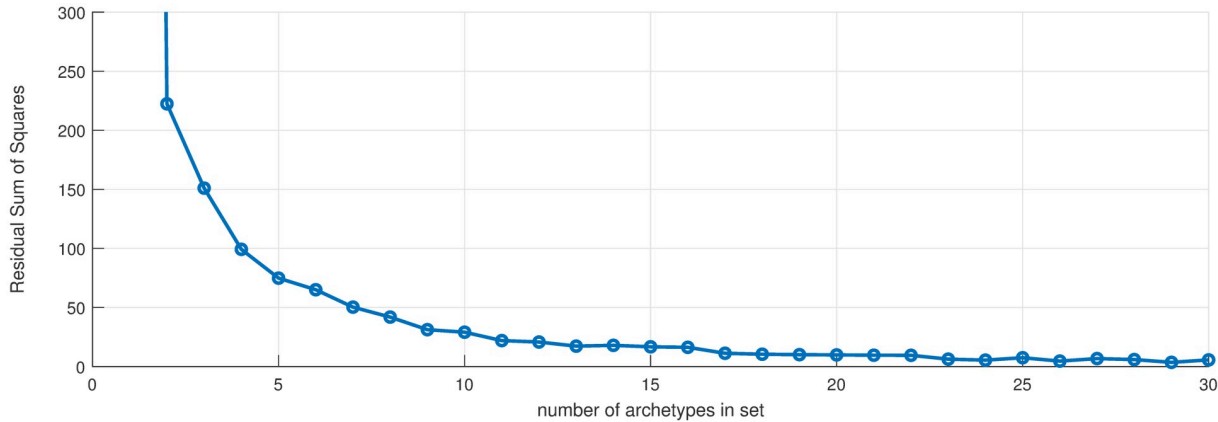

**Fig 7. Scree plot.** Residual sum of squares vs. number of archetypes being computed for the High MI County Set. Used to select the number of archetypes in the decomposition. Note: the RSS for 1 archetype is 3178, not seen with this *y*-range, which chosen to show the decay at larger numbers.

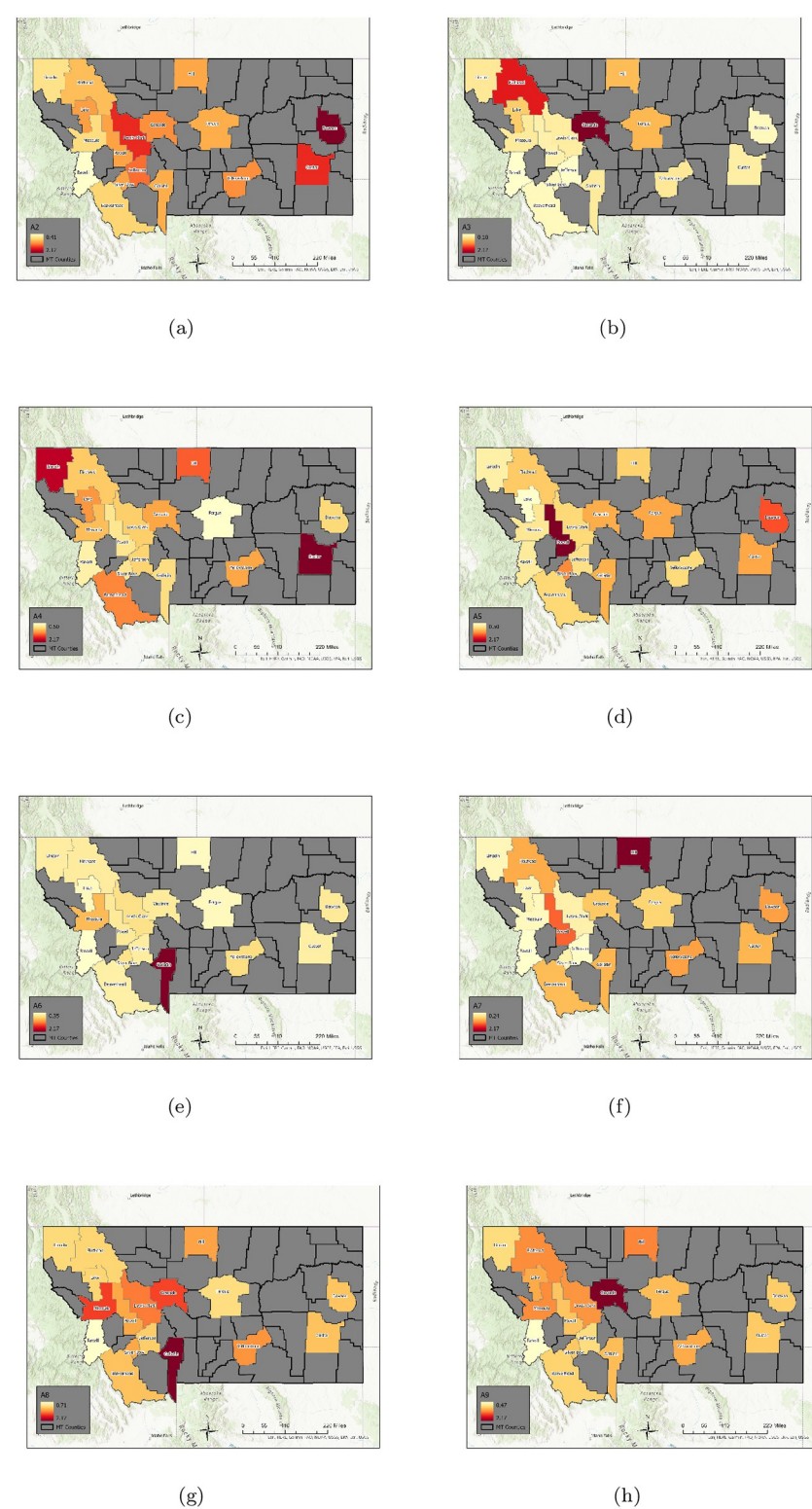

**Fig 8. Nine archetype set for the high MI county set.** Presented as color-coded counties in map of Montana. The first archetype is not included, as it captures the "no disease" state, and acts to "turn-off" the disease in each county. The rest are presented in order: [a] archetype 2, [b] archetype 3, [c] archetype 4, [d] archetype 5, [e] archetype 6, [f] archetype 7, [g] archetype 8, [h] archetype 9. Note that high $\beta$ values are considered those greater than 1.5, mid-size are between 0.75 and 1.5, low are below 0.75. Figure created with ArcGiS: Esri, HERE, Garmin, FAO, NOAA, USGS, EPA.

**Table 3. Archetype composition- largest MI county data set.**

| Archetype | Counties in Archetype |
|---|---|
| 1 | The zero or "no disease" archetype. |
| 2 | Widespread low outbreak across state. |
| 3 | Widespread low outbreak with large outbreaks in Cascade and Flathead counties. |
| 4 | Mid-size over all, larger in Lincoln, in the northwest, Hill in the north, and Custer in the east. |
| 5 | Mid-size outbreaks overall, with larger outbreaks in Powell and Dawson. |
| 6 | Low level outbreaks in all counties except in Gallatin, which is high. |
| 7 | Widespread outbreak, with lowest levels in Lincoln, Lake, Missoula, Ravalli, Lewis & Clark, Jefferson and Silverbow counties. Mid-size outbreaks in Flathead, Powell, Beaverhead, Cascade, Gallatin, Fergus, Yellowstone, Custer and Dawson. Larger outbreak in Hill. |
| 8 | Mid-size to very high level outbreaks over all. Largest in Gallatin, and large in Missoula, Lewis & Clark and Cascade counties. Powell and Silverbow have the next largest number of cases, then Yellowstone and Hill. The rest have more moderate size outbreaks. |
| 9 | Mid-size outbreaks in the contiguous region made up of Flathead, Lake, Missoula, and Lewis & Clark counties, with a large outbreak in adjacent Cascade county, and a mid-size outbreak in Hill county. The remaining counties have low to mid-size outbreaks. |

and 2, in that temporal order. The Early Delta low-level outbreak is captured by archetypes 6 and 3, the Delta outbreak by 4 and 2, and finally the Omicron outbreak is dominated by the sequence 6, 8, 9 and 3. There are small contributions from other archetypes as well, e.g. archetype 2 is a low widespread outbreak, and is present at the end of the first outbreak and the Delta outbreak. Archetype 6 and 3 are present to a lesser degree in the Omicron phase. These results suggest a different spatio-temporal spread during each phase, recapitulated in Table 4.

The analysis can be confirmed by comparing it to the time series of the major counties in each archetype during each phase. Fig 11 shows the COVID-19 time series in the counties with significant contributions to different archetypes, for comparison with the archetypal description above.

We see that COVID-19 initially appears in Hill county followed by Powell (represented by archetype 7), after which it spreads to a larger outbreak overall, with largest numbers in Powell, Silverbow and Dawson counties, represented by archetype 5. The last stage is characterized by large counts in Gallatin county, mid-size in Silverbow, Yellowstone, Lewis & Clark and Missoula counties, and smaller in the rest (seen in archetype 2).

The smaller Early Delta outbreak is captured by archetypes 2 6 and 3, with largest counts in Gallatin county followed by large counts in Cascade and Flathead counties.

In September the Delta outbreak continues, with archetype 4, switching to archetype 2. Archetype 4 has large $\beta$ values in Lincoln, Custer, Hill and Beaverhead counties, and archetype 2 is a widespread outbreak which is largest in Dawson county. We see in the time series that Lincoln and Custer do dominate initially, with Hill and Beaverhead also larger. As Lincoln and Custer counts decline, the Dawson count grows.

The Omicron outbreak was the most complex, as the counts grew and declined several times in different parts of the state, most likely reflecting the reduction in mitigation strategies combined with social gatherings, such as year-end holiday events, and the reconvening of schools in January. It outbreak began in December with archetype 6, switching to 8, then 9, then 3. This reflects the initial large case numbers in Gallatin county (archetype 6), followed by widespread outbreak with large counts also in Missoula, Cascade, and Lewis & Clark counties (archetype 8), then to large counts in Cascade (archetype 9) and finally decaying in all counties except Cascade and Flathead (archetype 3).

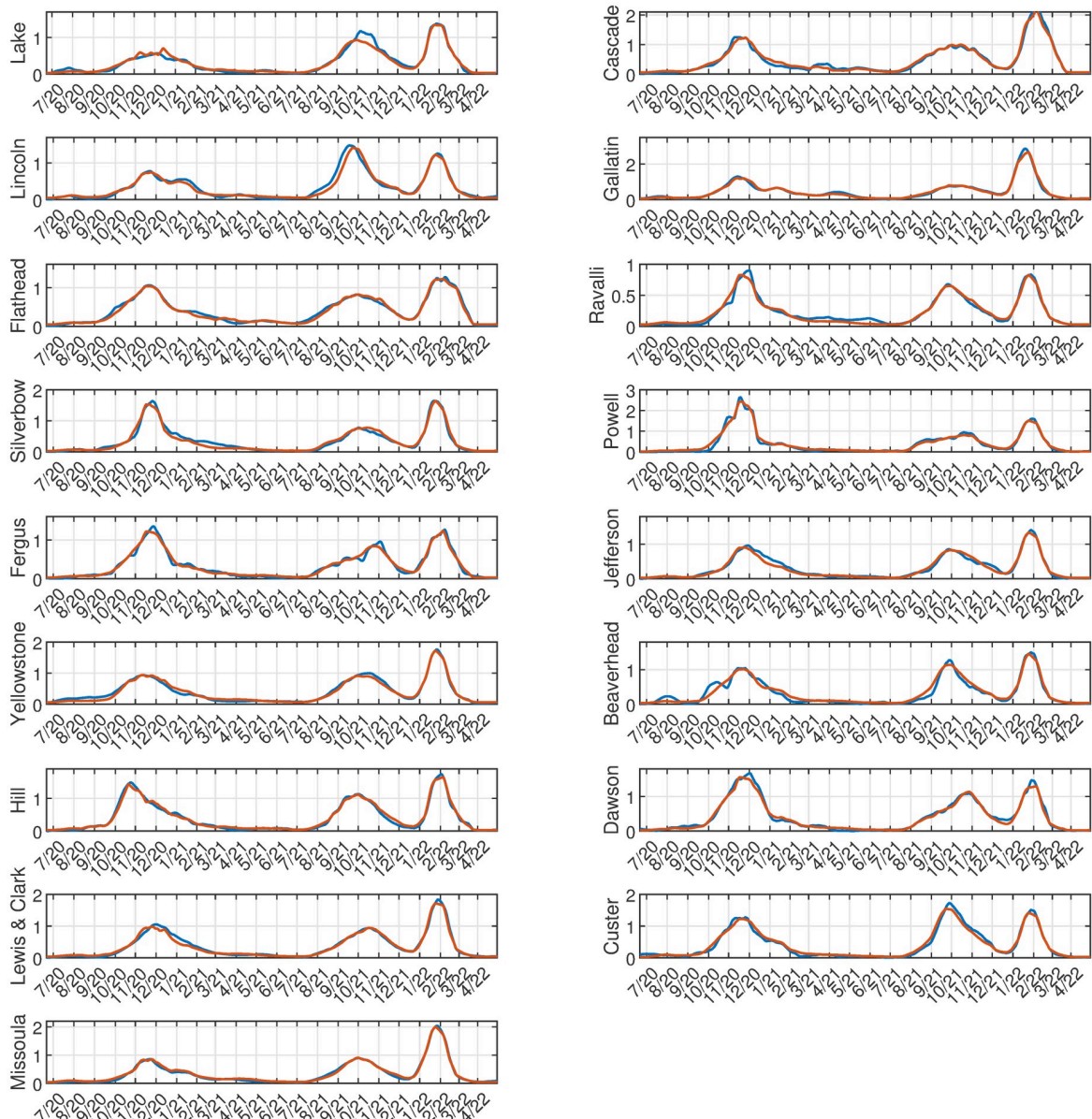

**Fig 9. Reconstruction of high MI county data set with 9 archetypes.** Original data are plotted in blue, and the reconstruction of the data in red.

## AA for the largest population counties

We next consider the counties with large population size to examine the spatio-temporal dynamics isolated to the cities and surrounding areas. There are 10 relatively large population cities in Montana, and we chose the counties that contained these cities. The time series for each is plotted in Fig 12.

For this analysis we choose a truncation to 6 archetypes, which captures roughly 98% of the variance, see Figs 13 and 14 shows color maps of each archetype, also described in Table 5.

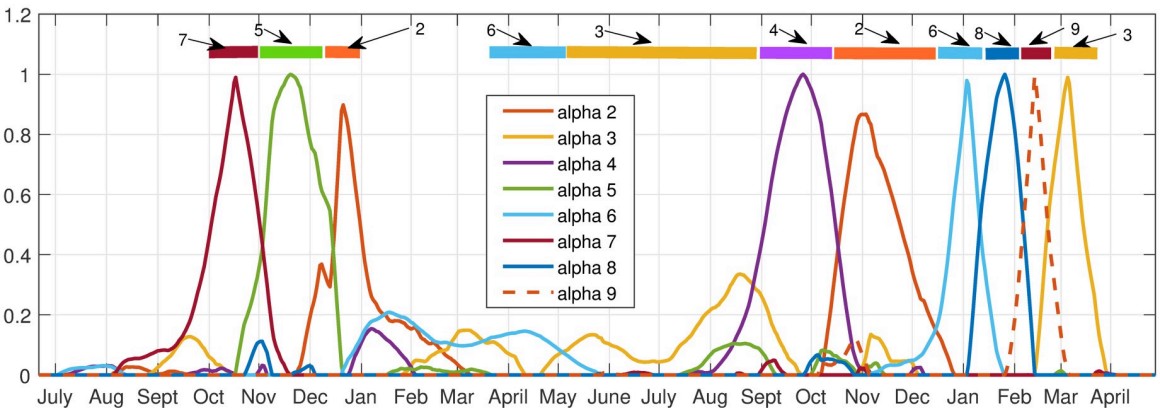

**Fig 10. Reconstruction coefficient ($\alpha$) time series for the 9 archetype decomposition of the 17 county data set.** Bars across top are color-coded to show the dominant archetype during that time period.

We can check the validity of the 6 archetype decomposition of the data by comparing the time series of the data for each county, and its reconstruction with archetypes. See Fig 15. The visible error is generally negligible, as expected because 98% of the variance of the data is captured with 6 archetypes.

As in the largest total MI data set, each archetype is almost uniquely identified with one of the major outbreaks. See Fig 16 in which the $\alpha$ time series is plotted along with colored bars that indicate the dominant archetype. Archetype 4 (Cascade, Flathead, Gallatin, Lewis & Clark, Silverbow) peaks during the Initial outbreak, archetype 2 (Cascade, Lake, Lewis & Clark, and Yellowstone) during the Delta outbreak. Archetypes 5 (Gallatin, Missoula), 6 (Cascade, Missoula, Lewis & Clark) and 3 (Cascade and Flathead) illustrate 3 different epochs in the Omicron phase. These archetypes also make contributions to the shoulders of the major outbreaks, and 3 and 5, at low levels, are used to represent the Early Delta outbreak. See Table 6.

We next compare the archetype time series with the data time series in Fig 17. In the initial outbreak, archetype 3 is first seen, because Gallatin, Cascade and Flathead counts rise first. In November 2020 archetype 4 becomes dominant, representing widespread contagion, with a large component in Silverbow county. As the outbreak declines overall in January, $\alpha_2$ and $\alpha_5$ rise, archetypes 2 and 5 giving a widespread lower level outbreak, with a larger component in Gallatin county.

The early Delta outbreak is represented by a low level of archetype 5 switching to a low level of archetype 3, giving a rise in Cascade county followed by a rise in Gallatin county, with slow spread into the other counties.

The Delta outbreak continues with low $\alpha_3$ transitioning to large $\alpha_2$, as it begins in Cascade and Flathead counties, then spreads to all counties with a larger component in Lincoln. Archetype 2 is thus the dominant archetype for the Delta outbreak. It ends as $\alpha_2$ declines and $\alpha_6$

**Table 4. Dominant archetypes for 17 high MI counties in each phase of the pandemic in montana, USA.**

| Phase/outbreak | 2 | 3 | 4 | 5 | 6 | 7 | 8 | 9 |
|---|---|---|---|---|---|---|---|---|
| Initial | ✓ | | | ✓ | | ✓ | | |
| Early Delta | | ✓ | | | ✓ | | | |
| Delta | ✓ | | ✓ | | | | | |
| Omicron | | ✓ | | | ✓ | | ✓ | ✓ |

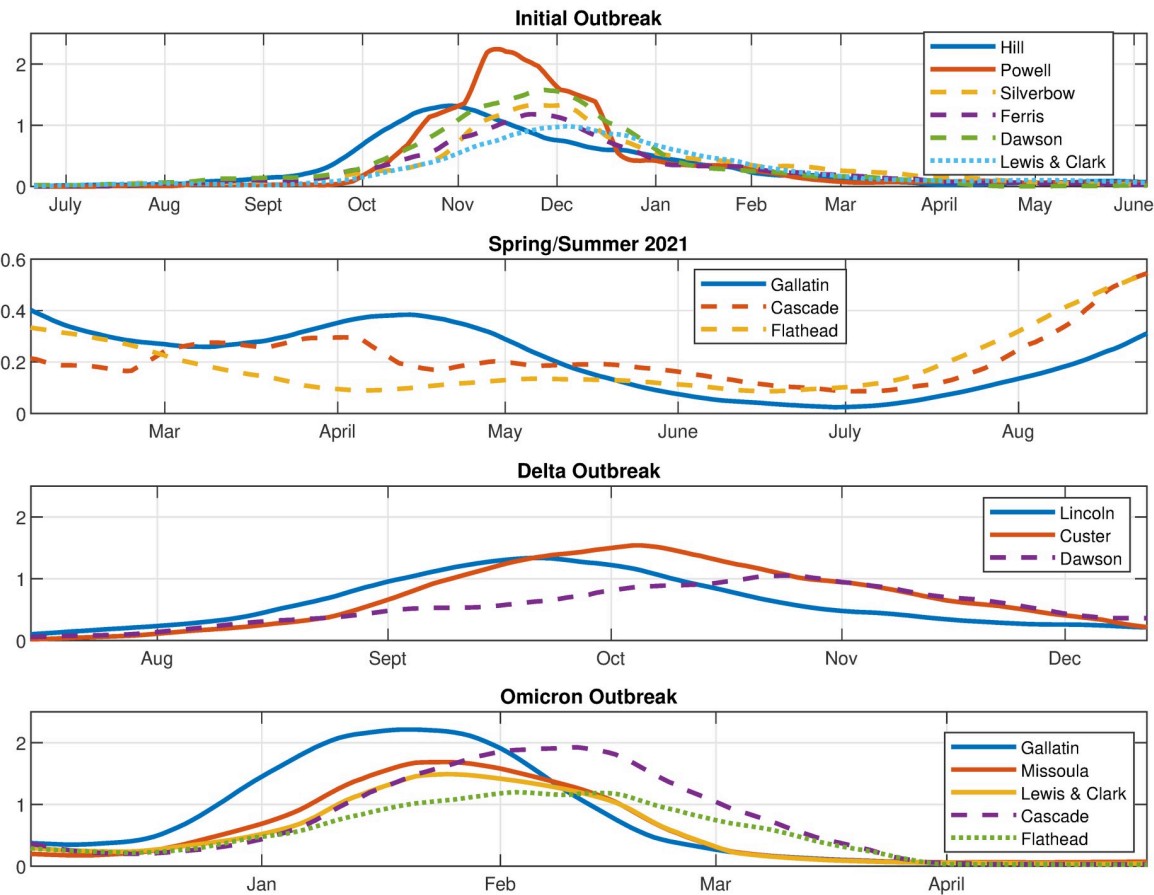

**Fig 11. Time series of major county components in the archetypes.** As labeled: Initial, Early Delta, Delta and Omicron outbreaks, 17 county data set.

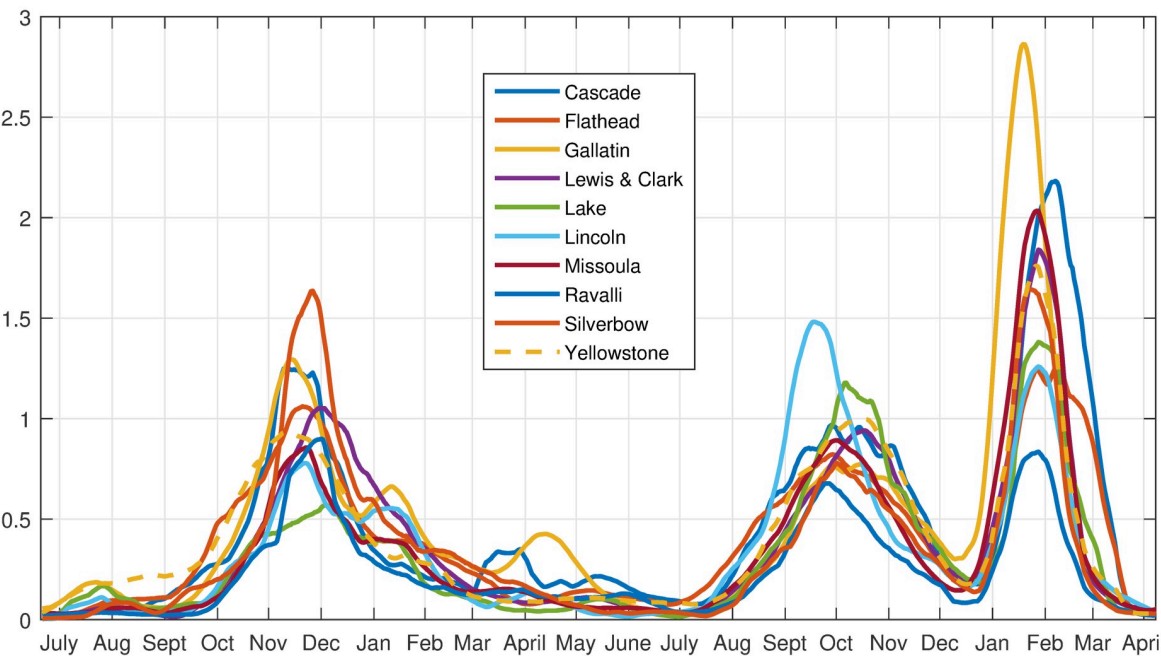

**Fig 12. Data time series: COVID case numbers for the large population county data set.**

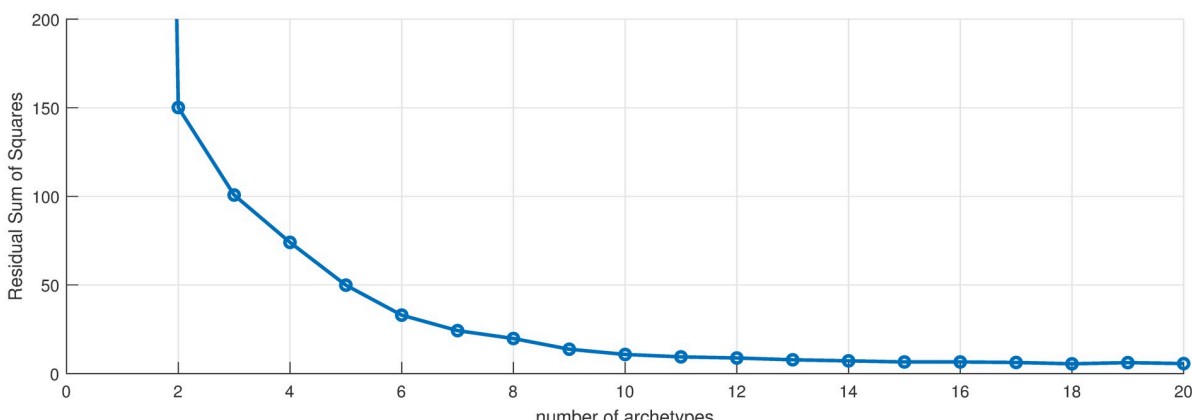

**Fig 13. Scree plot.** Residual sum of squares vs. number of archetypes for the Large Population County Set. Used to select the number of archetypes in the decomposition.

rises, reflecting the decline in Lincoln counts and the growing counts in all other counties, especially the counties with major cities, Missoula, Gallatin, Yellowstone, Cascade and Lewis & Clark.

The Omicron phase in late December 2021, begins with large counts in Gallatin (archetype 5) and switches to large counts in all other counties (archetype 6). It ends with significant counts in Cascade and Flathead, but low elsewhere, hence the rise of archetype 3.

## Discussion

We have shown how Archetypal Analysis can be used to good effect in studying a spatio-temporal data set of COVID-19 case counts in the counties of Montana, USA. Decomposing the entire 56 dimensional data set was problematic, however, because the large size of the per-population counts in small population counties form stochastic outliers in the data set and dominate the decomposition. To mitigate this, a straight-forward truncation to only large population counties with significant city centers formed one reduced set, but to include the small counties whose disease dynamics had significant interaction with the others, we used a mutual information (MI) measure between case count time series of the different counties, and chose those small counties with the largest MI, in addition to the large population counties, to create another set for analysis.

For each data set, the first archetype is necessarily the zero archetype, which indicates a disease free state in all counties. In the 17 county high total MI data set certain archetypes were tied uniquely to a given outbreak, while one archetype was used to represent low-level counts in between outbreaks. Including the small population counties showed the initiation of outbreaks from the boundaries of the state, in the north (Hill county), and the east (Custer county). Each of the archetypes of the 10 large population counties data set are also mostly identified with a single outbreak during the pandemic, and are similar to those in the 17 county set, restricted to the high population counties. For instance, archetype 4 for the 10 county data set is similar to archetype 5 in the 17 county data set (without the low population counties), and both are the main component of the first outbreak. In the Delta outbreak, archetype 2 from the 10 county set is similar to archetype 4 from the 17 county decomposition. In the Omicron outbreak, archetype 5 from the 10 county data set is similar to 8 from the 17

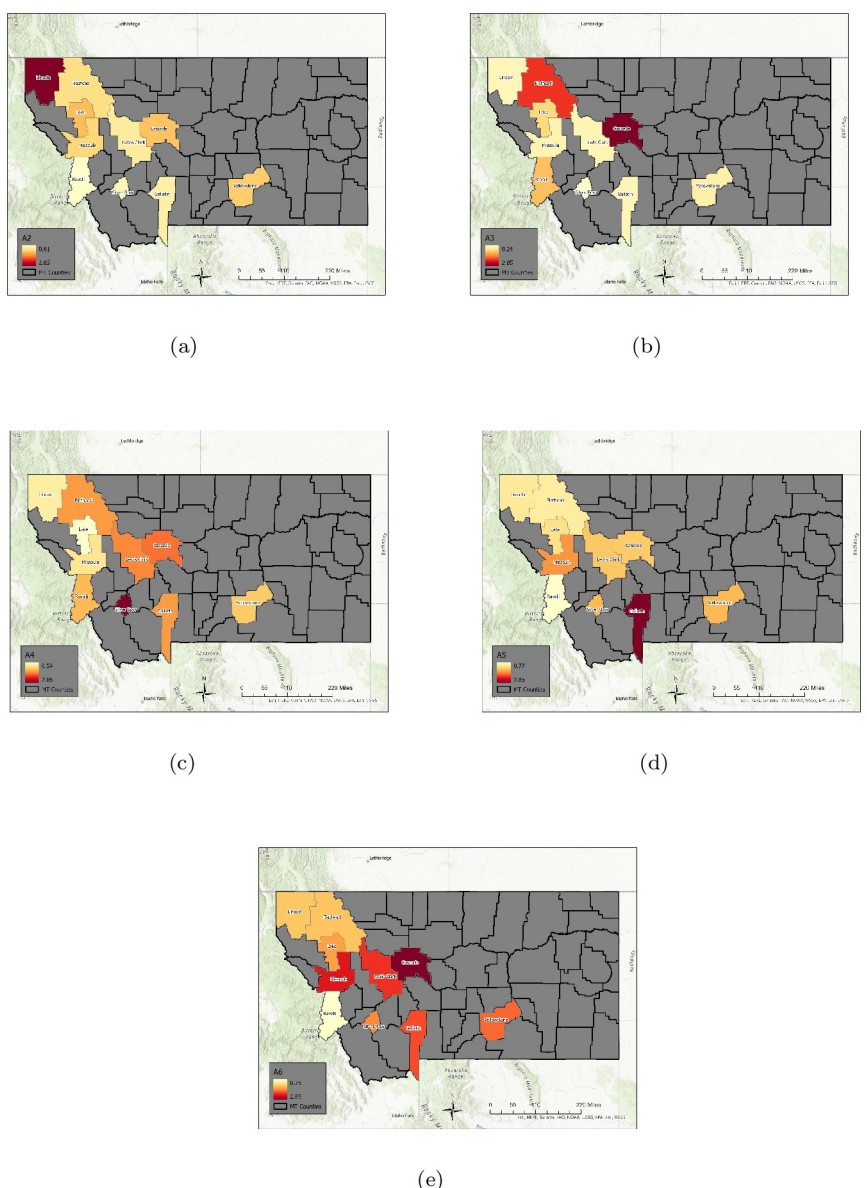

**Fig 14. The six archetypes for the large population county data set.** Note that the first archetype is not included, as it captures the "no disease" state, and acts to "turn-off" the outbreak in each county. The rest are in order: [a] archetype 2, [b] archetype 3, [c] archetype 4, [d] archetype 5, [e] archetype 6. Figure created with ArcGiS: Esri, HERE, Garmin, FAO, NOAA, USGS, EPA.

county data set, archetype 6 is similar to 9, and archetype 3 to 3. This overlapping structure is expected, and confirms the validity of the results.

The composition of an archetype itself can illustrate the geographic spread of the disease. For instance, Flathead, Lewis & Clark and Cascade counties have similar outbreak levels in several archetypes, which is not surprising, as they are linked by major state roads and have larger cities, but the analysis confirms that these connections are important. Other clusters of counties involved in outbreaks are revealed automatically in the archetypes, for instance, in the Initial outbreak a hot spot in Hill county and larger counts in Powell county then spread to all

**Table 5. Archetype composition-large population county data set.**

| Archetype | Counties represented in Archetype |
|---|---|
| 1 | The zero or "no disease" archetype. |
| 2 | Mid-size overall, with larger counts in Lincoln. |
| 3 | Low level outbreak overall, with mid-size outbreaks in Cascade and Flathead. |
| 4 | Mid-size outbreaks overall. Flathead, Lewis & Clark, Cascade and Gallatin form a set of counties with larger counts on a major transportation corridor. The largest counts are seen in Silver Bow county |
| 5 | Mid-size overall, with largest outbreak in Gallatin, followed by Missoula, Silver Bow and Yellowstone counties. Lewis & Clark and Cascade have the next highest level of outbreak, and the have low to mid-level sized outbreaks. |
| 6 | Large outbreaks in the west central counties of Cascade, Lewis & Clark and Missoula, with high mid-size outbreaks in the southern counties of Silver Bow, Gallatin and Yellowstone. The northwest corner counties of Lincoln, Flathead and Lake have mid-size outbreaks, with Ravalli county in the southwest corner with a low outbreak. |

eastern counties. Further analysis could be done by including on more counties in the east to determine how COVID-19 spreads west if initiated in the east. Archetypal decompositions could also be used to predict the spread of disease in future outbreaks. For instance, if an outbreak begins in a county that features largely in one archetype, would this imply spread to other counties with high levels in that archetype?

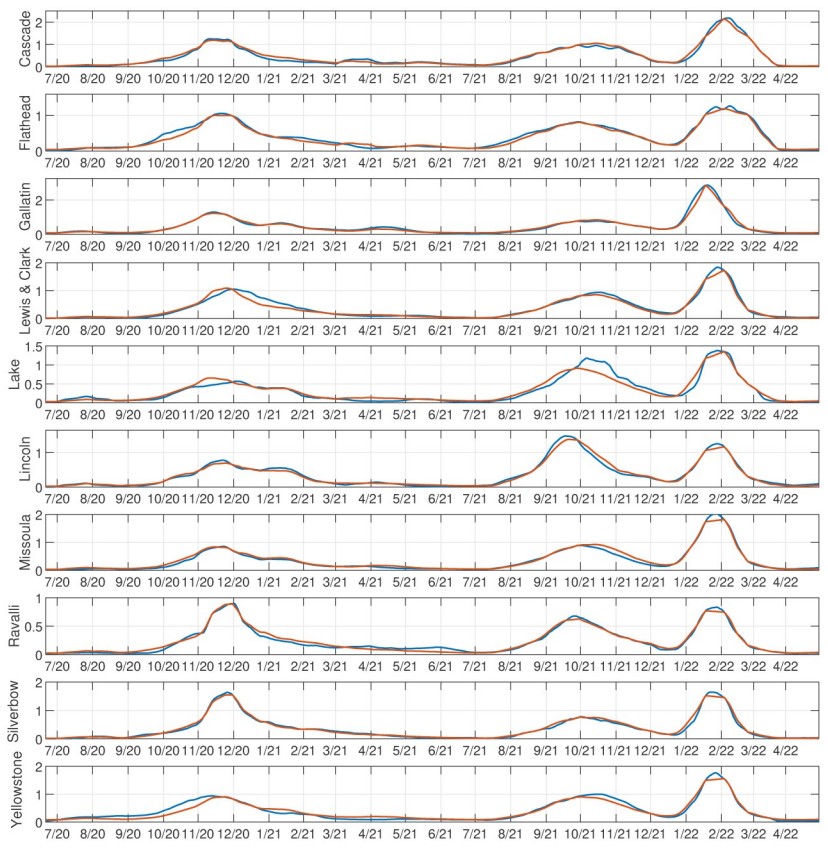

**Fig 15. Reconstruction of large population county data set with 6 archetypes.** Data are plotted in blue, reconstruction in red.

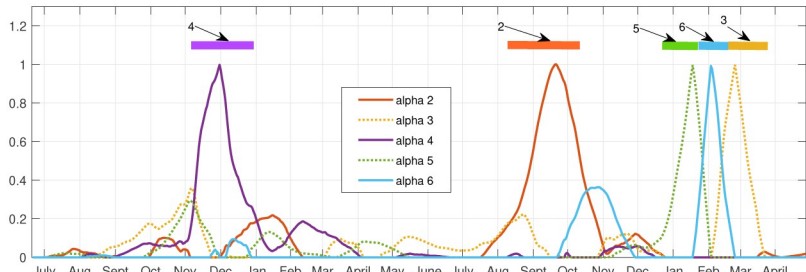

**Fig 16. Reconstruction coefficient ($\alpha$) time series.** The 6 archetype decomposition of the 10 county set. Bars across top are color-coded to show the dominant archetype during that time period.

**Table 6. Dominant archetypes for 10 large population counties in each outbreak of the pandemic in montana, USA.**

| Phase/outbreak | 2 | 3 | 4 | 5 | 6 |
|---|---|---|---|---|---|
| Initial | | | ✓ | | |
| Early Delta | | ✓ | | ✓ | |
| Delta | ✓ | | | | |
| Omicron | | ✓ | | ✓ | ✓ |

We close by commenting on the care that must be used in the creation and analysis of the archetypes. They have the advantage of automatically showing the counties that experience simultaneous outbreaks, and the state of the other counties during these time periods. Understanding the archetypes is intuitive, unlike graphical representations of principal component

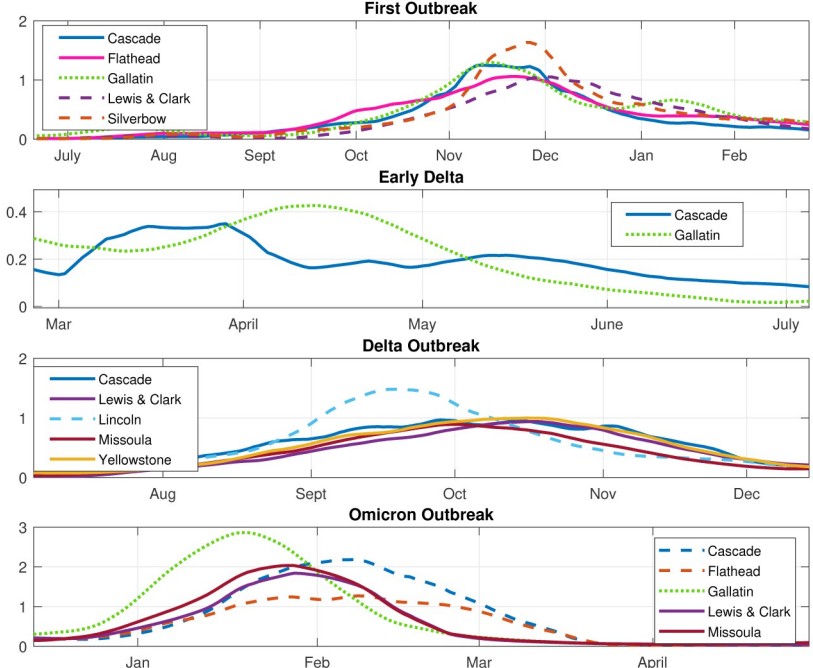

**Fig 17. Time series of major county components in the archetypes.** As labeled: Initial, early Delta, Delta and Omicron Phases, 10 county data set.

vectors. However, if outliers are present in the data (like the inflated counts from small population counties), they can bias the selection of the archetypes. This can be mitigated by filtering out these data points, as we did using an information measure between time series. In doing so we retained the small population counties that were important in the initiation of the outbreaks, which lay largely on the northern and eastern sides of the state. In conclusion, Archetypal Analysis, in tandem with human observation and intuition, can sift through complicated spatio-temporal data to reveal important features for further analysis of the phenomenon.

## Supporting information

**S1 File.**
(PDF)

## Acknowledgments

We thank the anonymous reviewers for offering feedback on manuscript. We also thank the Montana Department of Public Health and Human Services, Communicable Disease Epidemiology Section, for allowing us access to Montana's COVID-19 data.

## Author Contributions

**Conceptualization:** Emily Stone.

**Data curation:** Emily Stone, Erin Landguth.

**Formal analysis:** Emily Stone.

**Funding acquisition:** Erin Landguth.

**Investigation:** Emily Stone.

**Methodology:** Emily Stone.

**Project administration:** Emily Stone.

**Resources:** Emily Stone.

**Software:** Emily Stone.

**Supervision:** Emily Stone.

**Validation:** Emily Stone.

**Visualization:** Sebastian Coombs.

**Writing – original draft:** Emily Stone.

**Writing – review & editing:** Emily Stone, Erin Landguth.

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
