## [Decision Letter · Decision Letter 0]

8 Jun 2023

PONE-D-23-06211Archetypal analysis of COVID-19 in Montana, USA, March 13, 2020 to April 26, 2022PLOS ONE

Dear Dr. Stone,

Thank you for submitting your manuscript to PLOS ONE. After careful consideration, we feel that it has merit but does not fully meet PLOS ONE’s publication criteria as it currently stands. Therefore, we invite you to submit a revised version of the manuscript that addresses the points raised during the review process.

I strongly encourage the authors to address all the issues raised by both the reviewers. In particular, given your statement about the use of data publicly avaialble, I encourage you to share you data and code to allow a thorough check of the paper's analysis for the revision.

We look forward to receiving your revised manuscript.

Kind regards,

Maurizio Fiaschetti

Academic Editor

PLOS ONE

Journal Requirements:

2. Please note that PLOS ONE has specific guidelines on code sharing for submissions in which author-generated code underpins the findings in the manuscript. In these cases, all author-generated code must be made available without restrictions upon publication of the work. Please review our guidelines at https://journals.plos.org/plosone/s/materials-and-software-sharing#loc-sharing-code and ensure that your code is shared in a way that follows best practice and facilitates reproducibility and reuse

"This research was supported by the National Institute of General Medical Sciences of the National Institutes of Health (NIH), United States [Award Number P20GM130418]. We thank the anonymous reviewers for offering feedback on manuscript. We also thank he Montana Department of Public Health and Human Services, Communicable Disease Epidemiology Section, for allowing us access to Montana’s COVID-19 data. "

Please remove any funding-related text from the manuscript and let us know how you would like to update your Funding Statement. Currently, your Funding Statement reads as follows: "EL: This research was supported by the National Institute of General Medical Sciences of 428

the National Institutes of Health (NIH), United States [Award Number P20GM130418]

nigms.nih.gov

The funders did not play any role in study design, data collection and analysis, decision to publish, or preparation of the manuscript."

4. PLOS requires an ORCID iD for the corresponding author in Editorial Manager on papers submitted after December 6th, 2016. Please ensure that you have an ORCID iD and that it is validated in Editorial Manager. To do this, go to ‘Update my Information’ (in the upper left-hand corner of the main menu), and click on the Fetch/Validate link next to the ORCID field. This will take you to the ORCID site and allow you to create a new iD or authenticate a pre-existing iD in Editorial Manager. Please see the following video for instructions on linking an ORCID iD to your Editorial Manager account: https://www.youtube.com/watch?v=_xcclfuvtxQ.

6. We note that Figures 3a to 3j, 8a to 8h, 13a to 13e  in your submission contain [map/satellite] images which may be copyrighted. All PLOS content is published under the Creative Commons Attribution License (CC BY 4.0), which means that the manuscript, images, and Supporting Information files will be freely available online, and any third party is permitted to access, download, copy, distribute, and use these materials in any way, even commercially, with proper attribution. For these reasons, we cannot publish previously copyrighted maps or satellite images created using proprietary data, such as Google software (Google Maps, Street View, and Earth). For more information, see our copyright guidelines: http://journals.plos.org/plosone/s/licenses-and-copyright.

        1. You may seek permission from the original copyright holder of Figures 3a to 3j, 8a to 8h, 13a to 13e to publish the content specifically under the CC BY 4.0 license. 

Reviewers' comments:

Reviewer's Responses to Questions

**Comments to the Author**

1. Is the manuscript technically sound, and do the data support the conclusions?

Reviewer #1: Partly

Reviewer #2: Yes

2. Has the statistical analysis been performed appropriately and rigorously? 

Reviewer #1: No

Reviewer #2: Yes

3. Have the authors made all data underlying the findings in their manuscript fully available?

Reviewer #1: No

Reviewer #2: No

4. Is the manuscript presented in an intelligible fashion and written in standard English?

Reviewer #1: No

Reviewer #2: Yes

5. Review Comments to the Author

Reviewer #1: The paper describes an application of the archetypal analysis to study of COVID-19 epidemic in the state of Montana. While the application by itself is interesting from the methodological perspective, the manuscript has several important drawbacks:

1) The authors have not provided access to their data and code. This violates the community standards for data availability, which stipulates that data and code should be accessible to reviewers and readers of the paper for validation purposes.

2) The results are not clearly formulated, rendering the central message of the study unclear. The authors did not supply any quantitative or statistical evidence to substantiate the associations described in the Results section.

3) The rationale behind the abrupt shift from discussing COVID-19 pandemic results to flu on page 10 is unclear, as is the relevance of flu to the overall topic of the paper.

4) It is unclear why a significant portion of the results section (pages 5-6) is devoted to results acknowledged by the authors to be biased.

5) The decision to select 10 archetypes for the analysis (as stated on page 5) lacks a clear justification.

6) The paper is not well-written. For instance, the first paragraph of the introduction leaves readers questioning the essence of the "new implementations" that rendered spatio-temporal approaches tractable (implementations of what?), why these approaches were previously intractable, and the nature of the "patterns over time" mentioned. Furthermore, the sentence "the methods can detect spatial clustering could may reveal environmental causatives" is particularly opaque.

7) Table 1 is not cited in the paper.

Reviewer #2: The authors apply archival analysis (AA), an innovative tool similar to principal components analysis (PCA). The rationale for applying AA is convincing and the results are interesting.

Major Comment:

The subsection on the mathematical formulation has some problems. I believe these problems are in the explanation, not in any underlying issue with the method itself. The authors say “Given a specified value for k, AA identifies m-dimensional vectors z_1, z_2, … , z_k that best describe k characteristic patterns, or archetypes in the original data set … .” They go on to give some restrictions regarding these z’s, but they don’t say precisely what “best” means. The authors seem to make an attempt to make things precise at the bottom of p. 3 where they say

“The m x k matrix Z of k archetypes is defined by the matrix factorization problem:

min_{A,B} || X – XBA ||

where Z = XB.”

A couple of remarks on this statement. First, the authors don’t exactly say what the matrix factorization problem is. It actually looks like a minimization problem, not a matrix factorization problem. Is Z the minimum of || X – XBA || over all A and B? If so, they could write

Z = arg min_{A,B} || X – XBA ||.

I see no connection between Z and the minimization problem. Second, at the end of the sentence, they say Z = XB. So is Z the solution to the given minimization problem, or is it simply XB?

Minor Comments

1. The authors often talk about “observations” when they really mean “days.” It would help the reader to say there are data values for counties and days. Every time I saw “observations” I had to stop reading and think about what that means.

2. Near the bottom of p. 2, the authors talk about the AA algorithm. At this point of the paper, I didn’t have enough background to understand what any of this means. I’d suggest trimming this back, and then coming back to it later in the paper once the groundwork is laid.

3. It looks like the authors used Matlab, a package familiar to most mathematicians. Statisticians tend to use R because of its array of functions for doing data analysis. I believe there is an R package to do AA, which is called "archetypes". See the paper

Eugster, M., & Leisch, F. (2009). From spider-man to hero-archetypal analysis in R.

https://cran.r-project.org/web/packages/archetypes/vignettes/archetypes.pdf

4. On p. 3, first paragraph, the authors say “To account for differences in population size, we weighted the COVID-19 cases … .” “Weighted” could mean a lot of things in this context. Did the authors just look COVID rates on a per capita basis?

5. p. 3, just above equation (1). Should be “… coefficients that sum to unity …”

6. In equation (1) and thereafter, the betas are vectors and should be bold. This can be easily done in LaTeX using the amsmath package and the \\boldsymbol{} command.

7. p. 3, just after equation (1), Should be “Each archetype is either a convex combination of …”

8 p. 3, lines 84 and 90 (using numbering in the right margin). Write these matrices using brackets:

B = [ beta_1, … , beta_k ] and A = [ alpha_1 , … , alpha_n ]. These alphas and betas are themselves vectors and should be bold. Also use \\ldots for the ellipsis between commas; save \\cdots for the ellipsis between math operators like “+”.

9. In equation (1), the constraints must hold for all j. The “for all j” could be added a the end of (1).

10. The definition of mutual information at the bottom of p. 4 looked asymmetric to me. I got out some paper and tried to see whether it was symmetric, and of course it is. Then I turned the page and saw that the authors pointed out the symmetry. By the way, the description of MI at the top of p. 5 is very nice!

11. I couldn’t help but think that ten archetypes may be a bit much. I suspect that by the time you get past the first five or six, you get mostly noise. I’m not suggesting anything specific here, except maybe a statement that there will come a point beyond which we are observing noise.

12. p. 8, line 221. Shouldn’t “no flu” be “no COVID”? Same page, line 228, shouldn’t figure 8 be “Fig. 8”?

13. Throughout, I’d suggest not spelling out alpha. Just use $\\alpha$.

14. p. 13, lines 355-357. “A high (relative) MI indicates that a county’s time series can be better predicted by considering the other, and vice versa.” First, congratulations on getting “vice versa” correct; I always put a hyphen between the words, which is incorrect. I think the idea is that you can augment data from a county with other counties for which the MI is high. The way it is worded suggests that you ignore the data from the current county and just use the data from counties with a high MI.

15. The graphs are generally fuzzy. I know that PLOS ONE allows files in TIFF or EPS format only. EPS is a vector graphics format so the figures should be clear (no fuzziness at any level of magnification). I suspect the authors used TIFF files. The best strategy would be to render the graphs in EPS format. If this isn’t possible, then maybe saving the files at a higher TIFF resolution would make them clear.

16. In Figure 3, the axes are not needed. The legend on the right is, however, needed, but this can be given once (assuming the legend is the same for all archetypes). The figure will probably be set in a 5 by 2 array of maps so it fits on one page. The county names will not be readable at this scale. Similar comments apply to Figure 8. If the county names are needed, the authors might consider giving one map of Montana with just the county names. This figure could contain the map information (e.g., compass, scale in miles, surrounding topography, etc.) along with the county names.

6. PLOS authors have the option to publish the peer review history of their article (what does this mean?). If published, this will include your full peer review and any attached files.

Reviewer #1: No

Reviewer #2: No

---

## [Author Response · Author response to Decision Letter 0]

28 Sep 2023

To the Reviewers and the Editor:

We have extensively rewritten sections of the manuscript, to improve clarity and flow. The results have not changed, and the contents are almost identical to the first submission. Two figures and two tables have been added to address issues raised by the reviewers. 

The markup file shows the old submission, with the revisions added. Rewrites follow struck-out sections, and wholly new parts are highlighted in yellow.

We are not able to make the data publicly available, as it is comes from the MT Dept. of Health and Human Services. Readers can apply to them directly for access to the data. We did, however, put our code up on github, and the one package we used is listed in Supporting Information. 

Response to the Reviewers:

Reviewer #1: The paper describes an application of the archetypal analysis to study of COVID-19 epidemic in the state of Montana. While the application by itself is interesting from the methodological perspective, the manuscript has several important drawbacks:

1) The authors have not provided access to their data and code. This violates the community standards for data availability, which stipulates that data and code should be accessible to reviewers and readers of the paper for validation purposes.

We have added a copy of all code used in the supplementary information section, and have created a github repository of it, as well.

2) The results are not clearly formulated, rendering the central message of the study unclear. The authors did not supply any quantitative or statistical evidence to substantiate the associations described in the Results section.

We have added a clarification of the purpose of the study to the Introduction. We follow the lead of similar papers, such as Mørup and Hansen, 2010, that provide a proof of principle of applying archetypes to data from various sources. That said, the statistical fitness of the archetypes is built into the algorithm for finding them. The RSS of the truncation level chosen gives the total error of representing the data with that number of archetypes. We have included a new figure to show an approximation to the time series by the archetypes, to show this graphically. 

3) The rationale behind the abrupt shift from discussing COVID-19 pandemic results to flu on page 10 is unclear, as is the relevance of flu to the overall topic of the paper.

These are typos, thanks to the reviewer for spotting it. Should be COVID-19 or disease. It is changed. 

4) It is unclear why a significant portion of the results section (pages 5-6) is devoted to results acknowledged by the authors to be biased.

Another question that could be asked, is why not do archetypes on the entire data set? This section was included to explain the issues with doing do. We have highlighted the justification in the Abstract, Introduction and the Results section. 

5) The decision to select 10 archetypes for the analysis (as stated on page 5) lacks a clear justification.

We have highlighted the justification in the Abstract, Introduction and the Results section, where we state: “We next consider the counties with large population centers to examine the spatio-temporal dynamics isolated to the cities and their surrounding areas. There are 10 relatively large population cities in Montana, and we chose the counties that contained these cities.” 

6) The paper is not well-written. For instance, the first paragraph of the introduction leaves readers questioning the essence of the "new implementations" that rendered spatio-temporal approaches tractable (implementations of what?), why these approaches were previously intractable, and the nature of the "patterns over time" mentioned. Furthermore, the sentence "the methods can detect spatial clustering could may reveal environmental causatives" is particularly opaque.

The first paragraph has been rewritten and confusing statements reworded or removed.

7) Table 1 is not cited in the paper. 

The citation label was used twice, now fixed. Thanks to the reviewer for spotting this.

Reviewer #2: The authors apply archival analysis (AA), an innovative tool similar to principal components analysis (PCA). The rationale for applying AA is convincing and the results are interesting.

Major Comment:

The subsection on the mathematical formulation has some problems. I believe these problems are in the explanation, not in any underlying issue with the method itself. The authors say “Given a specified value for k, AA identifies m-dimensional vectors z_1, z_2, … , z_k that best describe k characteristic patterns, or archetypes in the original data set … .” They go on to give some restrictions regarding these z’s, but they don’t say precisely what “best” means. The authors seem to make an attempt to make things precise at the bottom of p. 3 where they say

“The m x k matrix Z of k archetypes is defined by the matrix factorization problem:

min_{A,B} || X – XBA ||

where Z = XB.”

A couple of remarks on this statement. First, the authors don’t exactly say what the matrix factorization problem is. It actually looks like a minimization problem, not a matrix factorization problem. Is Z the minimum of || X – XBA || over all A and B? If so, they could write

Z = arg min_{A,B} || X – XBA ||.

I see no connection between Z and the minimization problem. Second, at the end of the sentence, they say Z = XB. So is Z the solution to the given minimization problem, or is it simply XB?

We have clarified the notation in this section, and included “AA seeks to find $k$ $m$-dimensional archetypes such that the RSS is minimized”. That is the meaning of “best”.

Minor Comments

1. The authors often talk about “observations” when they really mean “days.” It would help the reader to say there are data values for counties and days. Every time I saw “observations” I had to stop reading and think about what that means.

Fixed.

2. Near the bottom of p. 2, the authors talk about the AA algorithm. At this point of the paper, I didn’t have enough background to understand what any of this means. I’d suggest trimming this back, and then coming back to it later in the paper once the groundwork is laid.

This paragraph has been moved to the Methods section

3. It looks like the authors used Matlab, a package familiar to most mathematicians. Statisticians tend to use R because of its array of functions for doing data analysis. I believe there is an R package to do AA, which is called "archetypes". See the paper

Eugster, M., & Leisch, F. (2009). From spider-man to hero-archetypal analysis in R.

https://cran.r-project.org/web/packages/archetypes/vignettes/archetypes.pdf

We added this reference to the Methods Section: From Spider-Man to Hero – Archetypal Analysis in R. Eugster MJA and Leisch F (2009). Journal of Statistical Software, 30(8), pp. 1–23.

4. On p. 3, first paragraph, the authors say “To account for differences in population size, we weighted the COVID-19 cases … .” “Weighted” could mean a lot of things in this context. Did the authors just look COVID rates on a per capita basis?

Changed the word to normalized, and yes, it makes the measurements per capita. 

5. p. 3, just above equation (1). Should be “… coefficients that sum to unity …”

Fixed.

6. In equation (1) and thereafter, the betas are vectors and should be bold. This can be easily done in LaTeX using the amsmath package and the \\boldsymbol{} command.

In equation 1 the beta is indexed by i and j and refers to a number. When beta is indexed by 1 subscript it is a vector. 

Following this recommendation, we made all the alpha and beta vectors appear in boldface.

7. p. 3, just after equation (1), Should be “Each archetype is either a convex combination of …”

Fixed

8 p. 3, lines 84 and 90 (using numbering in the right margin). Write these matrices using brackets:

B = [ beta_1, … , beta_k ] and A = [ alpha_1 , … , alpha_n ]. These alphas and betas are themselves vectors and should be bold. Also use \\ldots for the ellipsis between commas; save \\cdots for the ellipsis between math operators like “+”.

Fixed.

9. In equation (1), the constraints must hold for all j. The “for all j” could be added a the end of (1).

Fixed.

10. The definition of mutual information at the bottom of p. 4 looked asymmetric to me. I got out some paper and tried to see whether it was symmetric, and of course it is. Then I turned the page and saw that the authors pointed out the symmetry. By the way, the description of MI at the top of p. 5 is very nice!

Thank-you.

11. I couldn’t help but think that ten archetypes may be a bit much. I suspect that by the time you get past the first five or six, you get mostly noise. I’m not suggesting anything specific here, except maybe a statement that there will come a point beyond which we are observing noise.

The truncation level is always a tricky issue. There is an elbow in the scree plot around 6, but we found that the 6 archetype set missed some of the signal that we wanted to capture. Being able to check the reconstruction against the data allows for further refinement of the truncation. 

12. p. 8, line 221. Shouldn’t “no flu” be “no COVID”? Same page, line 228, shouldn’t figure 8 be “Fig. 8”?

We have removed all occurrences of the word “flu” in the document, and fixed the figure reference.

13. Throughout, I’d suggest not spelling out alpha. Just use $\\alpha$.

Fixed.

14. p. 13, lines 355-357. “A high (relative) MI indicates that a county’s time series can be better predicted by considering the other, and vice versa.” First, congratulations on getting “vice versa” correct; I always put a hyphen between the words, which is incorrect. I think the idea is that you can augment data from a county with other counties for which the MI is high. The way it is worded suggests that you ignore the data from the current county and just use the data from counties with a high MI.

Changed it to “A high (relative) MI between two counties, indicates that a county’s time series can be better predicted in tandem with the other’s, and vice versa.”

15. The graphs are generally fuzzy. I know that PLOS ONE allows files in TIFF or EPS format only. EPS is a vector graphics format so the figures should be clear (no fuzziness at any level of magnification). I suspect the authors used TIFF files. The best strategy would be to render the graphs in EPS format. If this isn’t possible, then maybe saving the files at a higher TIFF resolution would make them clear.

We used the PACE package to convert our .eps figures to .tiff, which is recommended by PLOS. From our end they look very clear. Not sure what the issue is with the draft the reviewer received, but we will make sure they are legible in the production copy.

16. In Figure 3, the axes are not needed. The legend on the right is, however, needed, but this can be given once (assuming the legend is the same for all archetypes). The figure will probably be set in a 5 by 2 array of maps so it fits on one page. The county names will not be readable at this scale. Similar comments apply to Figure 8. If the county names are needed, the authors might consider giving one map of Montana with just the county names. This figure could contain the map information (e.g., compass, scale in miles, surrounding topography, etc.) along with the county names.

Thanks to the reviewer for these helpful suggestions. 

---

## [Decision Letter · Decision Letter 1]

20 Oct 2023

Archetypal analysis of COVID-19 in Montana, USA, March 13, 2020 to April 26, 2022

PONE-D-23-06211R1

Dear Dr. Stone,

We’re pleased to inform you that your manuscript has been judged scientifically suitable for publication and will be formally accepted for publication once it meets all outstanding technical requirements.

Kind regards,

Maurizio Fiaschetti

Academic Editor

PLOS ONE

Additional Editor Comments (optional):

Reviewers' comments:

Reviewer's Responses to Questions

**Comments to the Author**

1. If the authors have adequately addressed your comments raised in a previous round of review and you feel that this manuscript is now acceptable for publication, you may indicate that here to bypass the “Comments to the Author” section, enter your conflict of interest statement in the “Confidential to Editor” section, and submit your "Accept" recommendation.

Reviewer #2: All comments have been addressed

Reviewer #3: All comments have been addressed

2. Is the manuscript technically sound, and do the data support the conclusions?

Reviewer #2: Yes

Reviewer #3: Yes

3. Has the statistical analysis been performed appropriately and rigorously? 

Reviewer #2: Yes

Reviewer #3: Yes

4. Have the authors made all data underlying the findings in their manuscript fully available?

Reviewer #2: No

Reviewer #3: Yes

5. Is the manuscript presented in an intelligible fashion and written in standard English?

Reviewer #2: Yes

Reviewer #3: Yes

6. Review Comments to the Author

Reviewer #2: (No Response)

Reviewer #3: The present study is interesting because it analyzes Archetypal to epidemiological data of COVID-19 from March 13, 2020 to April 26, 2022, for the counties of Montana, USA. The manuscript has already undergone an extensive review, with many details that have actually made the manuscript substantially improve its quality. The authors made the suggested adjustments appropriately and the manuscript can be accepted for publication.

7. PLOS authors have the option to publish the peer review history of their article (what does this mean?). If published, this will include your full peer review and any attached files.

Reviewer #2: **Yes: **Steven E. Rigdon

Reviewer #3: No

---

## [Editor Report · Acceptance letter]

24 Nov 2023

PONE-D-23-06211R1 

Archetypal analysis of COVID-19 in Montana, USA, March 13, 2020 to April 26, 2022 

Dear Dr. Stone:

I'm pleased to inform you that your manuscript has been deemed suitable for publication in PLOS ONE. Congratulations! Your manuscript is now with our production department. 

Kind regards, 

on behalf of

Dr. Maurizio Fiaschetti 

Academic Editor

PLOS ONE